

# Using cantor sets for error detection

Nithin Nagaraj

Consciousness Studies Programme, National Institute of Advanced Studies, Bengaluru, India

## ABSTRACT

Error detection is a fundamental need in most computer networks and communication systems in order to combat the effect of noise. Error detection techniques have also been incorporated with lossless data compression algorithms for transmission across communication networks. In this paper, we propose to incorporate a novel error detection scheme into a Shannon optimal lossless data compression algorithm known as Generalized Luröth Series (GLS) coding. GLS-coding is a generalization of the popular Arithmetic Coding which is an integral part of the JPEG2000 standard for still image compression. GLS-coding encodes the input message as a symbolic sequence on an appropriate 1D chaotic map Generalized Luröth Series (GLS) and the compressed file is obtained as the initial value by iterating backwards on the map. However, in the presence of noise, even small errors in the compressed file leads to catastrophic decoding errors owing to sensitive dependence on initial values, the hallmark of deterministic chaos. In this paper, we first show that repetition codes, the oldest and the most basic error correction and detection codes in literature, actually lie on a Cantor set with a fractal dimension of $\frac{1}{n}$, which is also the rate of the code. Inspired by this, we incorporate error detection capability to GLS-coding by ensuring that the compressed file (initial value on the chaotic map) lies on a Cantor set. Even a 1-bit error in the initial value will throw it outside the Cantor set, which can be detected while decoding. The rate of the code can be adjusted by the fractal dimension of the Cantor set, thereby controlling the error detection performance.

## INTRODUCTION

Computers and communication systems invariably have to deal with the ill effects of noise, which can lead to errors in computation and information processing. In a landmark paper published in 1950, Richard Hamming addressed this problem by introducing mathematical techniques for error detection and correction (*Hamming, 1950*). Since then, coding theory has burgeoned to be a field of its own right, boasting of important research and developments in the art and science of error detection and correction (*Lin & Costello, 1983*). Error detection/correction techniques have therefore been a fundamental part of most computing systems and communication networks, typically applied on the input data after data compression (lossy/lossless) and encryption (*Bose, 2008*).

Shannon's separation theorem (*Shannon, 1959*) states that under certain assumptions, data compression (source coding) and error protection (channel coding) can be performed

Corresponding author
Nithin Nagaraj, nithin@nias.res.in, nithin.nagaraj@gmail.com

[1]Progressive transmission refers to the successive approximation property of the bitstream where successive bits transmitted across the channel improve the fidelity of reconstruction at the decoder.

separately and independently (sequentially) while still maintaining optimality. However, in several practical scenarios, these assumptions are not met and hence there is a need for joint source channel coding. In image/video browsing and web-based applications, it is highly desirable to have the property of progressive transmission[1] (*Said & Pearlman, 1996*). However, even the slightest error can wreak havoc on a progressive bitstream. Hence, there is a need for incorporating error detection when decoding a compressed progressive bitstream.

Lossless data compression refers to the removal of redundancy in the data—a prerequisite step in most storage and communication systems before the application of an encryption algorithm and error control coding (for error detection/correction). Consider an i.i.d. binary message source $S$ which emits '0' with probability $p$ ($0 < p < 1$). A binary message $M$ of length $N$ from such a source needs to be losslessly compressed. Shannon (*Shannon, 1948*) showed that such a message can at best be losslessly compressed to $\geq H(S) \cdot N$ bits on an average, where $H(S)$ is the Shannon's entropy of the source. For an individual binary message $M$ from such an i.i.d. source, we can compute $H(\cdot)$ as follows:

$$H(M) = -p\log_2(p) - (1-p)\log_2(1-p) \quad bits/symbol, \tag{1}$$

where $p = \frac{Number\ of\ zeros\ in\ M}{N}$. There are several lossless compression algorithms in literature - Shannon-Fano coding, Huffman coding, Arithmetic coding, Lempel–Ziv coding and others (*Cover & Thomas, 2006*; *Sayood, 2000*). Among these, Arithmetic coding (*Rissanen & Langdon, 1979*) achieves the Shannon entropy limit for increasing message lengths. Arithmetic coding is used extensively in several practical applications owing to its speed, efficiency and progressive bitstream property. In fact, it is used in JPEG2000 (*Taubman & Marcellin, 2002*), the international standard for still image compression, replacing the popular Huffman coding which was used in the earlier JPEG (*Wallace, 1992*) standard.

In 2009 (*Nagaraj, Vaidya & Bhat, 2009*), it was shown that Arithmetic coding is closely related to a 1D non-linear chaotic dynamical system known as Generalized Luröth Series (GLS). Specifically, it was shown that lossless compression or encoding of the binary message $M$ can be performed as follows. First, the message $M$ is treated as a symbolic sequence on an *appropriately* chosen chaotic GLS. The initial value on the GLS corresponding to this symbolic sequence is computed by iterating backwards on the map. This initial value (written in binary) serves as the compressed file. For decompression, we start with this initial value (the compressed file) and iterate forwards on the (same) GLS, and record the symbolic sequence. This symbolic sequence is the decoded $M$. Such a simple lossless compression algorithm (known as GLS-coding) was proved to achieve Shannon's entropy limit (*Nagaraj, Vaidya & Bhat, 2009*). Arithmetic coding turns out to be a special case of GLS-coding (*Nagaraj, Vaidya & Bhat, 2009*).

Unfortunately, it turns out that GLS-coding (hence also arithmetic coding), having the progressive bitstream property, is very sensitive to errors. Even a single bit error in the compressed file can lead to catastrophic decoding errors. This has been well documented in the data compression literature (*Boyd et al., 1997*) and researchers have since been trying to enhance Arithmetic coding with error detection and correction properties (*Anand, Ramchandran & Kozintsev, 2001*; *Pettijohn, Hoffman & Sayood, 2001*). In this work, our

aim is to incorporate error detection into GLS-coding by using Cantor set while not sacrificing much on the lossless compression ratio performance of GLS-coding. As we shall demonstrate, Cantor set has desirable properties that enable efficient error detection with GLS-decoding.

The paper is organized as follows. In 'GLS-Coding and Decoding', we briefly describe GLS-coding of a binary message and its sensitivity to errors. In 'Error Detection Using Cantor Set', we explore Cantor set and describe its wonderful properties that enable error detection. Particularly, it is shown that Repetition codes belong to a Cantor set. Inspired by this, we incorporate error detection into GLS-coding using a Cantor set in 'Incorporating Error Detection into GLS-Coding Using a Cantor Set' and show simulation results of the proposed method. We conclude with open challenging problems in 'Conclusions and Future Work'.

## GLS-CODING AND DECODING

In this section, we shall briefly describe GLS-coding first proposed by *Nagaraj, Vaidya & Bhat (2009)*. We are given a message $M$, of length $L$, from a binary i.i.d. source $S$. Our aim is to losslessly compress $M$. To this end, we first determine the probability of zeros in $M$, denoted by $p$. We then construct the GLS map $T_p$ as follows:

$$x_{n+1} = T_p(x_n) = \begin{cases} \dfrac{x_n}{p}, & \text{if } 0 \leq x_n < p, \\ \dfrac{1 - x_n}{1 - p}, & \text{if } p \leq x_n < 1. \end{cases}$$

[2]One could alternately use the skew-binary map. There are eight possible *modes* of GLS in this case and all are equivalent in terms of compression ratio performance (*Nagaraj, Vaidya & Bhat, 2009*).

The GLS-map, also known as the skew-tent map[2], $T_p$, has two intervals: $[0, p)$ and $[p, 1)$ which are tagged with the symbols '0' and '1' respectively. Starting with any initial value $x_0 \in [0, 1)$, the above map $T_p$ can be applied iteratively to yield the real-valued time series $\{x_i\}$. The symbolic sequence of this time series, denoted by $\{S_i\}$, is simply obtained by:

$$S_i = \begin{cases} 0, & \text{if } 0 \leq x_i < p, \\ 1, & \text{if } p \leq x_i < 1. \end{cases}$$

Given the symbolic sequence $\{S_i\}$, the inverse of $T_p$ is given by:

$$T_p^{-1}(y_j) = \begin{cases} py_j, & \text{if } S_j = 0, \\ 1 - y_j(1 - p), & \text{if } S_j = 1, \end{cases}$$

For GLS-coding, we begin by treating the given message $M$ as a symbolic sequence $\{S_i\}_{i=0}^{i=L-1}$ of length $L$ on the GLS $T_p$. The goal is to obtain the initial value $x_0 \in [0, 1)$ which when iterated forwards on $T_p$ produces the symbolic sequence $\{S_i\}$ (= message $M$). To obtain $x_0$, we begin by initializing $START_0 = 0.0$ and $END_0 = 1.0$. We then determine the inverse images of $START_0$ and $END_0$ by iterating backwards on $T_p$, by using $T_p^{-1}$ and starting with the last symbol $S_{L-1}$. At the end of the first back-iteration, we obtain $START_1$ and $END_1$. We determine the inverse images of $START_1$ and $END_1$ (given $S_{L-2}$) to determine the new interval $[START_2, END_2)$. This is repeated $L$ times to yield the final interval $[START_{L-1}, END_{L-1})$. All points within this final interval have the same symbolic sequence

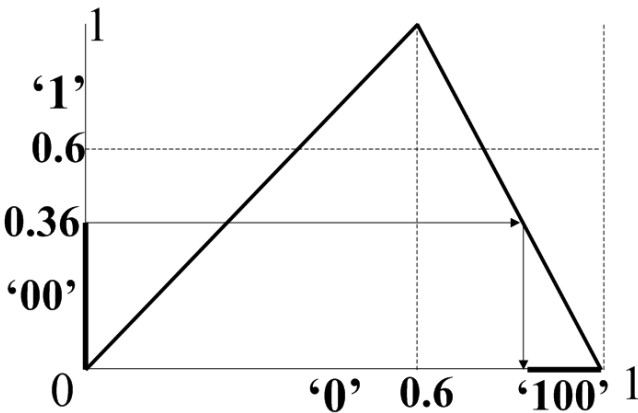

**Figure 1** **GLS-coding: an example.** For the message $M = 0010110\mathbf{100}$ ($L = 10$, p=0 .6), the above figure shows the backward-iteration from $\mathbf{00}$ to $\mathbf{100}$ on the GLS $T_{0.6}$. With $[START_0, END_0) = [0.0, 1.0)$ and iterating backwards, we obtain $[START_1, END_1) = [0.0, 0.6)$, $[START_2, END_2) = [0.0, 0.36)$, $[START_3, END_3) = [0.8560, 1.0)$ and so on. In this figure, when we go from the symbols $\mathbf{00}$ to $\mathbf{100}$, we go from $[START_2, END_2) = [0.0, 0.36)$ to $[START_3, END_3) = [0.8560, 1.0)$ since the symbolic sequence at this iteration is 1. This process is repeated until we have a final interval $[START_{L-1}, END_{L-1})$ for the entire $M$.

[3] The number of bits needed to store the compressed file $x_0$ approaches the entropy $H$ asymptotically as the length of the message approaches $\infty$.

[4] Arithmetic code turns out to be using the skewed-binary map instead of the skewed-tent map.

($= M = \{S_i\}$). We take the mid-point $x_0 = \frac{START_{L-1}+END_{L-1}}{2}$ as the compressed file. Since $x_0$ is a real number (between 0 and 1), we write its binary representation to the file. The number of bits needed to represent the compressed file $x_0$ is $\lceil -\log_2(END_{L-1} - START_{L-1}) \rceil$ bits. This is proved to be *Shannon optimal*[3] in *Nagaraj, Vaidya & Bhat (2009)* and Arithmetic coding is shown to be a special case of GLS-coding[4].

GLS-decoding is straightforward. At the decoder, given the value of $p$, we construct the GLS (skew-tent map) $T_p$ as described earlier. Given that we know $x_0$ (the compressed file), all that we need to do is iterate forwards on the map $T_p$ for $L$ ($=$ length of message $M$) iterations and output the symbolic sequence $\{S_i\}$. This is the decoded message and in the absence of any noise, this is exactly the same as $M$ which was input to the encoder.

As an example, consider $M = 0010110100$ ($L = 10$). In this case, $p = \frac{6}{10} = 0.6$ and $T_{0.6}$ is shown in Fig. 1. With $START_0 = 0.0$ and $END_0 = 1.0$, we obtain $[START_1, END_1) = [0.0, 0.6)$, $[START_2, END_2) = [0.0, 0.36)$, $[START_3, END_3) = [0.8560, 1.0)$ and so on.

### Effect of single-bit error on GLS-decoding: sensitive dependence on initial value of the map

Thus far, we have discussed lossless coding in the absence of any kind of noise. However, in practical applications, noise is unavoidable. If the data is in a compressed form, then it is quite likely that the decoder would be unable to decode or would decode incorrectly. Even detecting whether an error has occurred helps in several communication protocols since a repeat request could be initiated.

In GLS-coding, the compressed file is the *initial value* of the symbolic sequence (the message $M$) on the appropriate GLS. Since GLS is a chaotic map, it exhibits *sensitive dependence on initial values*, the hallmark of deterministic chaos (*Alligood, Sauer & Yorke, 1996*). A small perturbation in the *initial value* will result in a symbolic sequence which

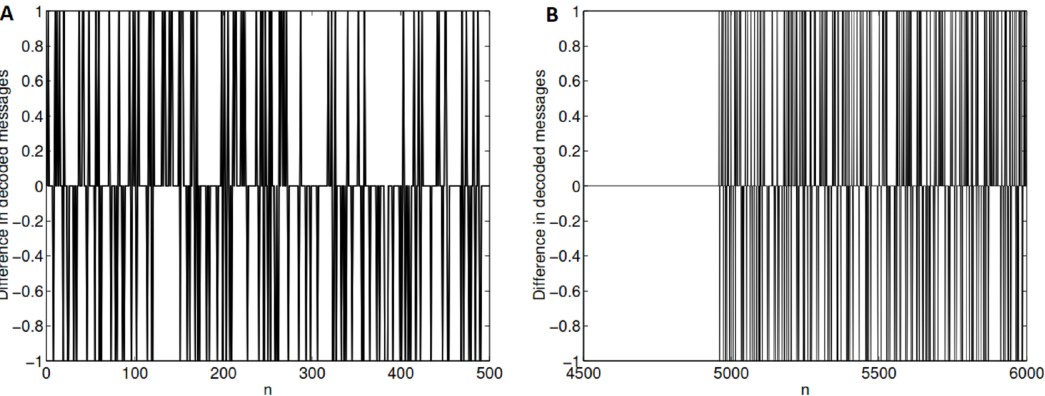

**Figure 2** **Effect of noise on GLS-coding.** (A) The first bit of the compressed file is flipped. The decoded message is very different from the actual intended message. (B) The middle bit (bit no. 3610) of the compressed file is flipped. The first 5,000 bits are decoded without error and the remaining 5,000 bits show lots of decoding error. ($p = 0.2$, $N = 10,000$, Compressed file size $= 7,220$ bits). In both cases, only part of the difference is shown.

is uncorrelated to the original symbolic sequence after a few iterations. This means that with a very high probability, even a slight amount of noise that is added to the initial value (compressed file) will result in a wrongly decoded message (symbolic sequence), which will be very different from the actual intended message. This is demonstrated in Fig. 2. The first bit of the compressed file is flipped and GLS-decoding is performed. The difference in the decoded message from the original message is shown in Fig. 2A. As it can be seen, the decoded message is very different from the original message. On the other hand, if the middle bit of the compressed file is flipped then the decoded image is accurate up to 5,000 bits and the remaining 5,000 bits are wrongly decoded (Fig. 2B). The error affects only those bits which are subsequently decoded.

## ERROR DETECTION USING CANTOR SET

In GLS-coding, every real number on the interval $[0, 1)$ represents an initial value (compressed file). Thus, any error in the initial value will result in another real number which is also an initial value, but for an entirely different symbolic sequence (message). It represents a valid compressed file which decodes to a different message. Therefore in order to detect errors, we necessarily require that when noise gets added to the initial value while transmission on the communication channel, it should result in a value that is not a valid compressed file, so that at the decoder it can be flagged for error. This necessarily implies that not all real numbers in the interval $[0, 1)$ can be valid compressed files. We need to restrict the set of valid compressed files to a smaller subset of $[0, 1)$. This subset should be uncountable and dense since it should be able to decode all possible (infinite length) messages. At the same time, it should have negligible measure (zero measure) so that when noise is added, the probability that it falls outside the set is 1. Cantor sets provide the perfect solution.

[5]The term 'Cantor Set' is used here in a more general sense to refer to fractals that are obtained very much like the middle-third Cantor set but with a different proportion (not necessarily $\frac{1}{3}$rd) removed at every iteration. Please see (*Strogatz, 2018*).

[6]Topological Cantor sets are not self-similar.

## The Cantor set

The well known middle-third Cantor set (*Alligood, Sauer & Yorke, 1996*; *Strogatz, 2018*) is a good example to illustrate this idea. All real numbers between 0 and 1 which do not have 1 in their ternary expansion belong to this Cantor set (call it $C$). We note down the following "paradoxical" aspects of Cantor sets[5] as observed in (*Strogatz, 2018*):

1. Cantor set $C$ is "totally disconnected". This means that $C$ contains only single points and no intervals. In this sense, all points in $C$ are well separated from each other.

2. On the other hand, $C$ contains no "isolated points". This means that every point in $C$ has a neighbor arbitrarily close by.

These two "paradoxical" aspects of Cantor sets (not just for the middle third Cantor set, but even for generalized Cantor sets, as well as, topological Cantor sets[6]) are actually very beneficial for error detection and correction. Property 1 implies that a small error will ensure that the resulting point is not in $C$ while Property 2 ensures that we can always find the nearest point in $C$ that can be decoded. Self-similar Cantor sets are fractal (their dimension is not an integer).

We shall show that repetition codes, one of the oldest error detection/correction codes lie on a Cantor set.

## Repetition codes $\mathcal{R}_n$ lie on a cantor set

Repetition codes are the oldest and most basic error detection and correction codes in coding theory. They are frequently used in applications where the cost and complexity of encoding and decoding are a primary concern. *Loskot & Beaulieu (2004)* provide a long list of practical applications of repetition codes. Repetition codes are robust against impulsive noise and used in retransmission protocols, spread spectrum systems, multicarrier systems, infrared communications, transmit delay diversity, BLAST signaling, rate-matching in cellular systems, and synchronization of ultrawideband systems[7]. Thus, repetition codes are very useful in communications.

[7]For further details, please see the references in *Loskot & Beaulieu (2004)*.

They are described as follows. Consider a message from a binary alphabet $\{0, 1\}$. A repetition code $\mathcal{R}_n$ ($n > 1$, odd integer) is a block code which assigns:

$$0 \mapsto \underbrace{0 \ldots 0}_{n}$$

$$1 \mapsto \underbrace{1 \ldots 1}_{n}.$$

$\mathcal{R}_n$ can correct up to $\frac{n-1}{2}$ bit errors since the minimum hamming distance of $\mathcal{R}_n$ is $n$. Since $n$ is chosen to be an odd positive integer ($> 1$), a majority count in every block of $n$ symbols acts as a very simple but efficient decoding algorithm. The repetition code $\mathcal{R}_n$ is a linear block code with a rate $= \frac{1}{n}$.

We shall provide a new interpretation of $\mathcal{R}_n$, inspired by Cantor set. Start with the real line segment $(0, 1]$. Remove the middle $(1 - 2^{-n+1})$ fraction of the set $(0, 1]$. In the remaining two intervals, remove the same fraction and repeat this process in a recursive fashion (refer to Fig. 3). When this process is carried over an infinite number of times, the set that remains is a Cantor set. Furthermore, the binary representation of every element of the Cantor set forms the codewords of $\mathcal{R}_n$.

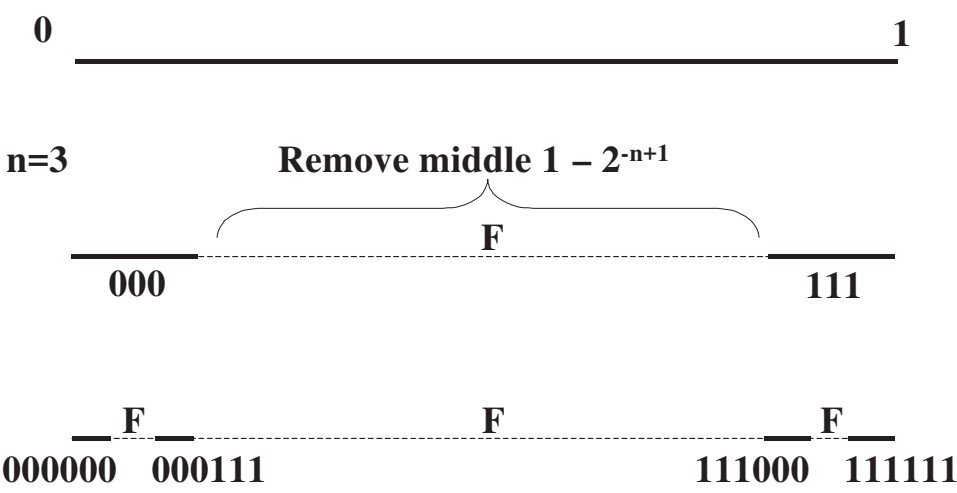

**Figure 3 Repetition codes $\mathcal{R}_n$ lie on a Cantor set: recursively remove the middle $(1 - 2^{-n+1})$ fraction.** As an example, $\mathcal{R}_3$ is depicted above. The box-counting dimension of the Cantor set is $D = \frac{1}{n}$. This is equal to the rate of the code.

In order to see this, consider $n = 3$. Figure 3 shows how $\mathcal{R}_3$ is recursively constructed. If the above step is terminated at iteration $k$, then there remains a set of intervals whose binary expansion (of length $nk$) contains the codewords for all possible binary messages of length $k$. For example, at $k = 2$ for $\mathcal{R}_3$, we can see that there are four intervals which contains real numbers with binary expansions starting from 000000, 000111, 111000 and 111111. These are the codewords for the messages 00, 01, 10 and 11 respectively. In the limit of this process, the set results in a Cantor set of measure zero which contains codewords for all binary messages which are infinitely long.

### Box-counting dimension of $\mathcal{R}_n$

We noted that repetition codes $\mathcal{R}_n$ lie on a Cantor set. It is very easy to compute the box-counting dimension of this Cantor set: $D = \lim_{\delta \to 0} \frac{\log B(\delta)}{\log(1/\delta)}$ where $B(\delta)$ is the number of boxes of size $\delta$ needed to cover the set. For $\mathcal{R}_n$, the box-counting dimension $D = \frac{1}{n}$ which is equal to the rate of the code. This establishes a very important connection between the properties of the Cantor set and the property of the code.

## INCORPORATING ERROR DETECTION INTO GLS-CODING USING A CANTOR SET

Having established that one of the oldest error detection/correction methods, namely repetition codes, belong to a Cantor set, we shall extend this idea of using a Cantor set for error detection to GLS-coding. First, we establish a connection between repetition codes and GLS-coding.

### Repetition codes re-visited

It is a common view to look at Repetition codes as block codes. In this section, we view them as GLS-coding with a forbidden symbol.

We could re-interpret Fig. 3 in a different way. Let the middle $1 - 2^{-n+1}$ interval be reserved for the forbidden symbol ' $F$ ' (this symbol never occurs in the message to be encoded) and the intervals $[0, 2^{-n})$ and $[1 - 2^{-n}, 1)$ correspond to the symbols '0' and '1' respectively. We treat all binary messages as symbolic sequences on this modified map and perform GLS-coding, i.e., find the initial value of a given message $M$. For GLS-coding, we are treating the alphabet $\{0, F, 1\}$ as taking the probabilities $\{2^{-n}, 1 - 2^{-n+1}, 2^{-n}\}$ respectively. The resulting initial value of GLS-coding is the codeword for the message and it turns out that it is the same as $\mathcal{R}_n(M)$. Thus, we have interpreted $\mathcal{R}_n(M)$ as a joint source channel code where the source has three alphabets and we are encoding messages that contain only 0 and 1.

By reserving a forbidden symbol '$F$' which is not used in encoding, all pre-images of the interval corresponding to '$F$' have to be removed. Thus, we have effectively created the same Cantor set that was referred to in the previous section. For error detection, one has to start with the initial value and iterate forwards on the modified map and record the symbolic sequence. If the symbolic sequence while decoding contains the symbol '$F$', then it invariably means that the initial value is not a part of the Cantor set and hence not a valid codeword of $\mathcal{R}_n$, thereby detecting that an error has occurred. Thus checking whether the initial value received belongs to the Cantor set or not is used for error detection at the decoder.

## GLS-coding with a forbidden symbol

We have presented two new ways of looking at Repetition codes—(1) the codewords of $\mathcal{R}_n$ lie on a Cantor set and (2) coding a message is the same as performing GLS-coding with a forbidden symbol reserved on the interval $[0, 1)$. The two are essentially the same because, by reserving a forbidden symbol $F$, we have effectively created a Cantor set on which all the codewords lie. But the fact that we can view $\mathcal{R}_n$ as GLS-codes enables us to see them as joint source channel codes for the source with alphabets $\{0, F, 1\}$ and with probability distribution $\{2^{-n}, 1 - 2^{-n+1}, 2^{-n}\}$ respectively. The natural question to ask is whether we can use the same method for a different probability distribution of 0 and 1. The answer is positive.

Instead of reserving a forbidden symbol $F$ of length $1 - 2^{-n+1}$, we could chose any arbitrary value $\epsilon > 0$ for the forbidden symbol. The value of $\epsilon$ determines the amount of redundancy that is going to be available for error detection/correction. It controls the fractal dimension of the Cantor set and hence the rate of the code. As $\epsilon$ increases, error detection/correction property improves at the cost of a slight reduction in compression ratio (note that the compression is still lossless, but no longer Shannon optimal). The probability of the symbol '0' is $p$, but only $(1 - \epsilon)p$ is allocated on the interval $[0, 1)$. Similarly, for the symbol '1': $(1 - \epsilon)(1 - p)$ is allocated. This single parameter $\epsilon$ can be tuned for trade-off between error control and lossless compression ratio. We shall show that a very small value of $\epsilon$ is sufficient for detecting errors without significantly increasing the compressed file size.

For encoding, as before, the binary message is treated as a symbolic sequence on the modified GLS with the forbidden symbol ' $F$ ' and the initial value is determined. The

initial value which is now on the Cantor set is the compressed file which is stored and/or transmitted to the decoder.

## Error detection with GLS-decoding

The decoder is the same as before except that we now have error detection capability. If the forbidden symbol ' $F$ ' is encountered during GLS decoding (this can happen only if noise corrupts the initial value/compressed file and throws it outside the Cantor set), then it is declared that an error has been detected. The decoder can then request the encoder to re-transmit the compressed file as is done in several protocols (*Chou & Ramchandran, 2000*). However, this scheme does not correct the error. It is quite possible that the noise is such that the initial value gets modified into another value which also happens to fall inside the Cantor set, in which case the decoder will not be able to detect the error (and thus we end up wrongly decoding the message). But, the probability of this occurring is very small (it is zero in the case of messages having infinite length since the measure of the Cantor set is zero). For finite length messages, the probability of such an event is given by the measure of the set of codewords (which is non-zero). In the following section, we perform rigorous experimental tests of the proposed approach.

## Simulation results

Three different values of $\epsilon$ ($\epsilon_1 = 0.005$, $\epsilon_2 = 0.03$, $\epsilon_3 = 0.05$) for the length of the interval corresponding to the forbidden symbol ' $F$ ' were used. The amount of redundancy that needs to be added is easy to determine. By introducing the forbidden symbol of length $\epsilon$, the valid symbols occupy a sub-interval of length $1 - \epsilon$. Thus, each time a symbol with probability $p$ is encoded, $-\log_2((1-\epsilon)p)$ bits will be spent, whereas only $-\log_2(p)$ bits would have been spent without the forbidden symbol. Thus, the amount of redundancy is $R(\epsilon) = -\log_2((1-\epsilon)p) + \log_2(p) = -\log_2(1-\epsilon)$ bits/symbol. For $N$ symbols, this would be $N \cdot R(\epsilon)$ bits rounded to the nearest highest integer. Thus the rate of the code will be:

$$\text{Rate} = \frac{1}{1 + R(\epsilon)} = \frac{1}{1 - \log_2(1-\epsilon)}. \tag{2}$$

As expected, this is equal to the box-counting dimension of the Cantor set. Thus, by plugging in $\epsilon = 1 - 2^{-n+1}$ in Eq. (2), we obtain the rate of the repetition codes as $\frac{1}{n}$.

We introduced a single bit error (one bit is flipped in the entire compressed file) towards the end of the compressed file for binary i.i.d. sources ($p = 0.1, 0.3$). Note that, it is much more difficult to detect errors if they happen towards the end of the compressed file than if it occurred in the beginning of the file. This is because, any error can only affect decoding for subsequent bits and if the error was towards the end-of-file (EoF), not many bits are available to catch it. Remember that the Cantor set (having zero measure) is obtained only after an infinite number of iterations. Since we are terminating after a finite number of iterations, we don't strictly get a Cantor set. In other words, the set that remains when we terminate after a finite number of iterations is an approximation to the Cantor set and it contains points which would have been discarded if we had continued iterating. The single-bit errors closer to the EoF thus decode to points which are not discarded because of this approximation (as we run out of iterations). These errors survive and go undetected.

**Table 1 GLS-coding with forbidden symbol: redundancy ($N = 10{,}000$, CFS, Compressed File Size).**

| $p$ | CFS $\epsilon = 0$ (bits) | $\epsilon_1 = 0.005$ | | $\epsilon_2 = 0.03$ | | $\epsilon_3 = 0.05$ | |
|---|---|---|---|---|---|---|---|
| | | $N \cdot R(\epsilon_1)$ (bits) | CFS (bits) | $N \cdot R(\epsilon_2)$ (bits) | CFS (bits) | $N \cdot R(\epsilon_3)$ (bits) | CFS (bits) |
| 0.1 | 4,690 | 72 | 4,762 | 440 | 5,130 | 740 | 5,430 |
| 0.3 | 8,812 | 72 | 8,881 | 440 | 9,253 | 740 | 95,52 |

**Table 2 GLS-decoding with forbidden symbol: error detection ($p = 0.1$).** EoF stands for 'End-of-File'.

| Location of single bit-error introduced | Number of error events | $\epsilon_1 = 0.005$ | | $\epsilon_2 = 0.03$ | | $\epsilon_3 = 0.05$ | |
|---|---|---|---|---|---|---|---|
| | | Detected | Undetected | Detected | Undetected | Detected | Undetected |
| EoF to EoF −49 | 50 | 16 | 34 | 31 | 19 | 41 | 9 |
| EoF −50 to EoF −99 | 50 | 18 | 32 | 48 | 2 | 49 | 1 |
| EoF −100 to EoF −149 | 50 | 32 | 18 | 49 | 1 | 50 | 0 |
| EoF −150 to EoF −249 | 100 | 92 | 8 | 100 | 0 | 100 | 0 |
| Total | 250 | 158 | 92 | 228 | 22 | 240 | 10 |

**Table 3 GLS-decoding with forbidden symbol: error detection ($p = 0.3$).** EoF stands for 'End-of-File'.

| Location of single bit-error introduced | Number of error events | $\epsilon_1 = 0.005$ | | $\epsilon_2 = 0.03$ | | $\epsilon_3 = 0.05$ | |
|---|---|---|---|---|---|---|---|
| | | Detected | Undetected | Detected | Undetected | Detected | Undetected |
| EoF to EoF −49 | 50 | 9 | 41 | 27 | 23 | 36 | 14 |
| EoF −50 to EoF −99 | 50 | 15 | 35 | 43 | 7 | 48 | 2 |
| EoF −100 to EoF −149 | 50 | 29 | 21 | 49 | 1 | 50 | 0 |
| EoF −150 to EoF −249 | 100 | 69 | 31 | 99 | 1 | 100 | 0 |
| Total | 250 | 122 | 128 | 218 | 32 | 234 | 16 |

In our simulations, the location of the single bit error was varied from the last bit to the 250th bit from end-of-file. Thus, the total number of single-bit error events introduced in the compressed bitstream is 250 for each setting of $\epsilon$. This way, we can test the proposed method under the worst scenario[8].

Table 1 shows the amount of redundancy owing to the allocation of the forbidden symbol. Tables 2 and 3 shows the performance of the method for $p = 0.1$ and $p = 0.3$. As expected, higher values of $\epsilon$ are able to detect more errors, but at the cost of increased compressed file size. Table 4 shows the efficiency of the method. Up to 96% of single bit errors introduced at the tail of the compressed file are detected by a modest increase in the redundancy (up to 15.78%). It should be noted that errors introduced in the beginning of the compressed file can be very easily detected by the proposed method.

[8] It should be noted that errors introduced in the beginning of the compressed bitstream were always detected.

**Table 4  GLS-coding with forbidden symbol: efficiency.**

| $p$ | % of errors detected | | |
|---|---|---|---|
| | $\epsilon_1 = 0.005$ | $\epsilon_2 = 0.03$ | $\epsilon_3 = 0.05$ |
| 0.1 | 63.2% | 91.2% | 96.0% |
| 0.3 | 48.8% | 87.2% | 93.6% |

### Arithmetic coding with a forbidden symbol: prior work

The idea of using a forbidden symbol into arithmetic coding was first introduced by *Boyd et al. (1997)*. It was subsequently studied by *Chou & Ramchandran (2000)*, *Grangetto & Cosman (2002)*, *Anand, Ramchandran & Kozintsev (2001)*, *Pettijohn, Hoffman & Sayood (2001)* and *Bi, Hoffman & Sayood (2006)*. However, the approach that is taken in this paper is unique and entirely motivated by a non-linear dynamical systems approach, through the wonderful properties of Cantor sets. We are thus able to justify why the method actually works. To the best of our knowledge, none of the earlier researchers have made this close connection between error detection/correction for repetition codes or arithmetic coding and Cantor sets. This work paves the way for future research on error correction using fractals/Cantor sets and potentially a host of new efficient techniques using Cantor sets could be designed.

## CONCLUSIONS AND FUTURE WORK

In this work, we have provided a novel application of Cantor sets for incorporating error detection into a lossless data compression algorithm (GLS-coding). Cantor sets have paradoxical properties that enable error detection and correction. Repetition codes are an example of codewords on a self-similar Cantor set which can detect and correct errors. By reserving a forbidden symbol on the interval $[0, 1)$, we can ensure that the codewords for GLS-coding lie on a Cantor set and thereby detect errors while simultaneously GLS-decoding (thus preserving progressive transmission property), and without significantly increasing the compressed file size. This approach can be applied to any mode of GLS and generalizable to larger alphabets. However, we do not know whether other efficient error control codes can be similarly designed using such Cantor sets (or other fractals in higher dimensions) and whether we can exploit the structure of the Cantor set to perform efficient error correction. These are important research directions worth exploring in the future.

## ACKNOWLEDGEMENTS

The author expresses sincere thanks to Prabhakar G. Vaidya for introducing him to the fascinating field of Non-linear dynamics/Chaos, Cantor sets and Fractals, and to William A. Pearlman for introducing him to the equally exciting field of data compression.

### Funding

This work was supported by Tata Trusts. The funders had no role in study design, data collection and analysis, decision to publish, or preparation of the manuscript.

### Grant Disclosures

The following grant information was disclosed by the author:
Tata Trusts.

### Competing Interests

There are no competing interests.

### Author Contributions

- Nithin Nagaraj conceived and designed the experiments, performed the experiments, analyzed the data, contributed reagents/materials/analysis tools, prepared figures and/or tables, performed the computation work, authored or reviewed drafts of the paper, approved the final draft.

### Data Availability

MATLAB code is available in a Supplemental File.

### Supplemental Information

Supplemental information for this article can be found online at http://dx.doi.org/10.7717/peerj-cs.171#supplemental-information.

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
