# Peer review of "Using cantor sets for error detection"

_PeerJ Computer Science, doi:10.7717/peerj-cs.171_

## Round 0.1 · original submission · Minor Revisions

The reviewers have made excellent suggestion that will help improve the quality and contents of the paper. I look forward to the revised version.

Reviewer 1 ·

Basic reporting

Few sentences can be rephrased::

ine 58:: demonstrate, Cantor set has desirable properties that enable efficient error detection while<with> GLS-decoding.

line 61:: that enable error detection. Repetition codes are shown to lie on a Cantor set<sentence can be framed in a better way>. Inspired by this, we incorporate error detection into GLS-coding using a Cantor set in section 4 and show simulation results of the proposed method. We conclude with open challenging problems in section 5.

< Para can be re-phrased>
line 159 Repetition codes which are error detection/correction codes lie on a Cantor set. How can we extend this
160 idea of placing codewords on a Cantor set for GLS-coding? Here, we establish the connection between
161 repetition codes and GLS-coding.

<Section heading can be re-phrased>
201 4.3 Error Detection while<with> GLS-Decoding

line 202 : If while GLS decoding, the forbidden symbol ‘F’ is encountered
can be rephrased as :
<If the forbidden symbol 'F' is encountered during GLS decoding>


line 215: The amount of redundancy that is added is easy to determine.
can be rephrased to:
<The amount of redundancy that needs to be added is easy to determine>

line 226: this line can be re-phrased
This is because, any error can only affect decoding for subsequent bits and if the error was towards the end-of-file (EoF), not many bits are available to catch it.

Experimental design

4.4 Simulation Results
The author did not mention the number of iterations, confidence interval

Validity of the findings

no comment

Additional comments

1) The novelty this paper is in identifying that Repetition codes are closely related to cantor sets. The author has done only preliminary experiments.
Is it possible to do experiments on real workloads.
What are the advantages/disadvantages of using this scheme?
Under what conditions this scheme performs better? Under what conditions it does not perform well?


2) Can the author also estimate (quantify) the probability of undetected errors - they mention that it is non-zero for finite length messages.

3) Looks like the error model - here is a 1 bit error. They did not explicitly mention this.

4) Can the author compare the current technique with other existing techniques which provide the same amount of detection capability. For e.g parity can also detect single-bit errors. So, why should anyone use this particular approach if parity is better.

Reviewer 2 ·

Basic reporting

The description of the author's prior work on GLS coding is needlessly succinct. Unless there are page limit restrictions that need to be satisfied It should be expanded to at least include an example. The idea is straightforward enough that it really should not need much additional work.


The use of a forbidden symbol was also used in Pettijohn et al. (2001) and should probably be included in section 4.5.

Experimental design

no comment

Validity of the findings

no comment

Additional comments

Nice work!

Reviewer 3 ·

Basic reporting

The reporting is basically sound, but certain clarifications would be helpful to the reader.
The map from source symbol to interval could be better illustrated and explained.
START and END should be labeled or indicated in Fig. 1 or its caption. Initial value corresponding to '001' and number of bits to send it should be given.
Why is reserving a width epsilon for a forbidden symbol ensure codewords lying on a Cantor set? I thought the Cantor set remains when middle third is repetitively removed.
The properties of "totally disconnected" and "isolated points" are true of the Cantor set, where the middle third is repetitively removed ad infinitum. Clearly, that is impractical and sub-intervals containing points of the Cantor set remain. So a single bit error may not take the value outside the sub-interval in which it occurs. The error might be detected if follow a sufficient number of sub-divisions of that interval. I think an explanation along these lines would better clarify the reason why a later bit error may not be caught if not enough bits remain in the code sequence.

Experimental design

I have no comments. All aspects are in order.

Validity of the findings

The experimental findings look to be correct and support well the validity and efficacy of the method to detect bit errors. I am somewhat confused by the first column in Tables 2 and 3. This column gives ranges of distances of a single bit error from EOF. Since there is only one corresponding number for Detected and Undetected, it seems that these numbers count the Detected and Undetected errors in that range; and the bit error is inserted prior to the range's beginning point. If so, I suggest a change of column heading wording. Otherwise, everything else is clear.

Additional comments

The paper is well-written and presents a novel scheme for continuous detection of transmission errors using GLS decoding. Please take into account the above comments. One additional minor matter:
in 3.2, you mention correction of up to (n-1)/2 bit errors. Only true when n is odd, or change to
floor( (n-1)/2 ).

---

## Round 0.2 · accepted · Accept

After reviewing your response, I believe that you have responded to reviewers comments adequately.